# Whole Exome-Trio Analysis Reveals Rare Variants Associated with Congenital Pouch Colon

**DOI:** 10.3390/children10050902

**Published:** 2023-05-19

**Authors:** Sonal Gupta, Praveen Mathur, Ashwani Kumar Mishra, Krishna Mohan Medicherla, Obul Reddy Bandapalli, Prashanth Suravajhala

**Affiliations:** 1Department of Biotechnology and Bioinformatics, Birla Institute of Scientific Research (BISR), Statue Circle, Jaipur 302021, India; 2Amity Institute of Biotechnology, Amity University Rajasthan, Kant Kalwar, Jaipur 303002, India; 3Department of Pediatric Surgery, SMS Medical College and Hospital, JLN Marg, Jaipur 302004, India; 4DNA Xperts, Noida 201301, India; 5Department of Bioengineering, Birla Institute of Technology, Mesra, Jaipur Campus, 27-Malaviya Industrial, Area, Jaipur 302017, India; 6Bioclues.org, Hyderabad 500072, India; bandapalli@gmail.com; 7Amrita School of Biotechnology, Amrita University, Vallikavu, Clappana P.O. Box 690525, Kerala, India

**Keywords:** whole exome sequencing, trio exome, missense variants, congenital pouch colon

## Abstract

Anorectal malformations (ARM) are individually common, but Congenital Pouch Colon (CPC) is a rare anorectal anomaly that causes a dilated pouch and communication with the genitourinary tract. In this work, we attempted to identify de novo heterozygous missense variants, and further discovered variants of unknown significance (VUS) which could provide insights into CPC manifestation. From whole exome sequencing (WES) performed earlier, the trio exomes were analyzed from those who were admitted to J.K. Lon Hospital, SMS Medical College, Jaipur, India, between 2011 and 2017. The proband exomes were compared with the unaffected sibling/family members, and we sought to ask whether any variants of significant interest were associated with the CPC manifestation. The WES data from a total of 64 samples including 16 affected neonates (11 male and 5 female) with their parents and unaffected siblings were used for the study. We examined the role of rare allelic variation associated with CPC in a 16 proband/parent trio family, comparing the mutations to those of their unaffected parents/siblings. We also performed RNA-Seq as a pilot to find whether or not the genes harboring these mutations were differentially expressed. Our study revealed extremely rare variants, viz., *TAF1B*, *MUC5B* and *FRG1*, which were further validated for disease-causing mutations associated with CPC, further closing the gaps of surgery by bringing intervention in therapies.

## 1. Introduction

Whole Exome Sequencing (WES) has been an invaluable and cost-effective approach to identify genetic variants responsible for both Mendelian and polygenic diseases [1]. In the recent past, it has allowed the detection of clinically relevant genomic regions spanning the known unknown regions, disease-associated sites and untranslated regions (UTRs) [2]. In addition to the well-known diseases, prenatal abnormalities, structural anomalies and congenital defects were studied, demonstrating a good diagnostic yield [3,4]. While WES approaches are abundant, they are limited if the disease in question is characteristically rare and medically inconclusive. This could be a deterrent because of the challenges in variant discovery, including rare and low-frequency mutations using next generation sequencing (NGS) technologies. A recent decrease in the cost of WES and the accuracy of the NGS enabled the researchers to study a large number of WES samples, but in case of rare diseases, studying exome-trios (proband/parents), or quads, with an addition of siblings, to discover single nucleotide variations (SNVs) and indels has proved to be a major landmark in the discovery of rare disease variants. For example, Zhang et al., 2021; Gabriel et al., 2022; and Jiang et al., 2021 employed characteristic trio-exome analysis to infer the candidate or driver mutation in rare diseases [5,6,7]. With human disease and genetic variation studies largely driven by NGS, a paramount challenge would be to explore de novo mutations, i.e., those not inherited from either the father or mother. To check this, parent–child trios/quad WES analysis could be a powerful approach, although biases impede the identification of potential de novo mutations. Nevertheless, a majority of mutations may not transcend from parent to offspring, making it necessary to comprehensively analyze genetic variants in order to confirming them as associated [8]. Trio-based exome sequencing has provided beneficial for identifying de novo variants in rare diseases, attributing them to largely heterozygous/causal mutations [9]. For rare diseases, although WES analysis often makes assumptions regarding disease inheritance (de novo vs. recessive), variant frequency and genetic heterogeneity, it has opened the path towards improved disease management or prognosis and effective therapies. For example, WES trios in schizophrenia patients for recessive genotypes were studied with rare mutations in voltage-gated sodium ion channels contributing to the disorder [10]. In another study, Jin et al. (2017) identified pathogenic mutations with an increased rate of de novo mutations in early-onset high myopia (EOHM) patients [11]. Recently, Quinlan-Jones et al. correlated the proband–parent trios to determine the clinical utility of molecular autopsy underlying the etiology of structural anomalies [12]. In addition, through whole-exome sequencing, Hu et al. examined complete genetic variants including de novo variants with rare sporadic cases of non-syndromic hearing loss [13]. Similarly, trio-based whole exome sequencing was applied on cell-free fetal DNA, and revealed a de novo frameshift variant of the X-linked *STAG2* gene [14].

Congenital Pouch Colon (CPC) is a rare type of high anorectal malformation wherein a part of or the entire colon becomes dilated in the form of a pouch with a fistula connecting genitor–urinary tract [15]. Most have been reported in India with cases common to other countries accounted for, although males are prone to be largely affected, with a male to female ratio of 4:1 [16]. The incidence of CPC is highest in north-west regions of India, and is estimated to be 5–18% of the total number of neonates managed for anorectal malformations [15]. From WES approaches, we have earlier identified mutations that are associated with CPC and reported candidate missense mutations [17]. In another study, we also inferred the role of long non-coding RNAs (lncRNAs) from a WES and identified lnc-EPB41-1-1, located in the intergenic regions of EPB41, which is known to interact with KIF13A [18]. In this extended WES study of CPC, we examined the role of rare allelic variations associated with CPC in a 16 proband/parent trio family, comparing the mutations to those of their unaffected parents/siblings. Keeping in view the understanding that the genetic basis of CPC could possibly delve into variation, an attempt was made to discover variants contributing to phenotypic heterogeneity. In the present study, we initially performed genomic analysis using WES to screen the causal variants associated with CPC [17], and then identified potential contributing rare variants in novel/plausible candidate genes. For this, we investigated the parental origins of probable disease-causing rare variants using whole exome trio analysis.

## 2. Materials and Methods

### 2.1. Trio Selection, Samples Collection and Ethical Approval

The CPC subjects were recruited from the J.K. Lon Hospital, SMS Medical College, Jaipur, India, in accordance with a protocol approved by the institutional ethics committee (IEC) of the hospital. Written informed consent was provided by the parents on behalf of their children. Blood samples were collected from all the probands, parents and unaffected siblings, if any. The WES data from a total of 64 samples, including 16 affected neonates with their parents and unaffected siblings, were used for the study. We confined our pool of analyses to all probands (11 male and 5 female) and unaffected parents/siblings. The methods and pipeline leading to family/quad analyses are summarized in Figure 1 (Appendix A).

### 2.2. Variant Annotation, Filtering and Quality Control

The details of sequencing and variant calling in CPC subjects have been described previously [17]. Briefly, WES was performed on an Illumina multiplexed sequencer with paired-end chemistry and 110x effective coverage. Using our in-house developed pipeline [19], all unmapped sequence reads were aligned to the human reference genome (hg38) and variants were called. The mutations from WES study were checked to discover the variants, if any, across the samples (Figure 2). All variants with MAF < 0.05 were checked for whether or not they were present in probands but absent in their respective parents (Appendix A) and healthy siblings. After checking the variants, we confined the prioritization of variants with filters set to an average depth of 250 and MAF < 0.01 and MAF ≤ 0.01% across all the trio samples [20]. For further checking with dbSNP [21], GnomAD [22], ClinVar [23], and COSMIC [24] databases, we used SNP-Nexus [25] to filter mutations listed in a cohort of databases, viz., SIFT [26], PolyPhen-2 [27], Ensembl Variant Effect Predictor [28], MutationTaster [29], CADD [30] and GERP [31], and prioritized the pathogenic mutations, if they were deleterious in nature. The CNVs and variants of unknown significance (VUS) were inferred by mapping the final list of variants to SNP-Nexus. As a final check in reaching a consensus for the variants present in all the probands, we checked the variants with multiple bioinformatics tools so as to find bona fide variants at the union of intersection of these methods, which we construed to be associated with CPC.

### 2.3. Downstream Analyses

We used the vcftools package (https://vcftools.github.io; last accessed on 6 March 2023), and the mutations in probands were further checked with their relative parents/siblings for heterozygosity transmission. We also looked into the homozygous variants and considered them as associated with CPC, where the proband was found to be homozygous and their respective parents and unaffected siblings, if any, were heterozygous for the specific allele. Enrichment analysis of the data was carried out to calculate the inclusiveness of parameters such as binomial probability and hypergeometric distribution [32]. After high throughput screening, we undertook candidate gene-set analysis based upon significantly enriched sets or rare mutations, specifically in colon related disorders. Seeking novel insights into the disorder, we used pathway analysis based upon gene ontology (GO) derived from PANTHER ontology [33] (http://pantherdb.org/tools; last accessed on 6 March 2023) and EnrichR [34] (https://amp.pharm.mssm.edu/Enrichr; last accessed on 6 March 2023) annotation terms. Gene Ontology-based annotations included a biological network gene ontology tool (BinGO) [35], a plug-in for ontology annotation in Cytoscape [36] used for ontological analysis in the form of biological, cellular and metabolic processes.

### 2.4. Identification of Transcripts for Comparative Screening

From RNA-Seq, a quality check ensued after total RNA was isolated and after cDNA double strand synthesis on a pair of CPC type-4 samples. The RNA-Seq was performed on an Illumina HiSeq 2000 platform with 2 × 100 bp paired end sequencing chemistry which generated ca. 32 million read pairs. The pair of samples was run through differential gene expression analysis using Cufflinks [37] and DESeq [38] pipelines and a consensus was reached. For inferring the role of lncRNAs, UVA FASTA software (https://fasta.bioch.virginia.edu/fasta_www2/fasta_list2.shtml; last accessed on 6 March 2023) (v36. 6.8 version) and the NONCODE FASTA repository [39] were downloaded and the intergenic regions of the genes from WES samples were queried. The lncRNA—NONHSAT002007 was identified based on the query coverage e-value < 0.01. The sequences were carefully checked for bidirectional blast hits, and the lncRNA was visualized using an Ensembl genome browser for fidelity.

### 2.5. SNP Genotyping and Burden Tests

For Sequenom genotyping, a multiplexed iPLEX assay was designed for 20 ng of DNA per sample to determine SNP calls using the Agena biosciences assay design suite. The MALDI-TOF-MS analysis was performed using the Agena biosciences massArray analyzer platform on 29 SNPs in 37 DNA samples (16 probands and 21 controls). This method consisted of five steps: PCR amplification, shrimp alkaline phosphatase treatment, single base extension, nano dispensing, and matrix-assisted laser desorption/ionization–time of flight (MALDI-TOF) mass spectrometry [40]. Data acquisition was automatically performed, and the mass window of analyte peak observation was set at 4500–9000 Da. Call frequencies, expressed as percentages, were calculated for each SNP. The final list of bona fide variants’ mutational burden was checked by comparing the previous analysis [17]. All the variants reported were carefully taken into consideration through the iterative process of the American College of Medical Genetics and Genomics (ACMG) [41] and Sherloc guidelines [42].

## 3. Results and Discussions

### 3.1. Variant Downstream Analysis

We achieved a mean read depth of 80 in the targeted regions for an average depth of coverage of 110x. We generated an average of 840,667 total variants, comprising 775,262 SNPs and 65,405 indels per exome in probands, as compared to 736,820 SNPs and 63,418 indels in unaffected samples from a total of 777,216 variants (Figure 3). Missense variants constituted the large variation, followed by loss of function variants in cases when compared to unaffected samples, with overall 0.35% missense, 0.02% frameshift, 0.004% stop gain and 0.0009% stop lost. The downstream analysis leading to the variant calling was carried out carefully to yield the list of the final number of variants. This was also checked with gene-density and high linkage disequilibrium (LD) regions even as MAF ≤ 0.01 was sought for, with SNPsnap [43] having no candidate matches, thereby confirming that these variants are extremely rare. Given the rare phenotype, the frequency of the prioritized variants was checked for agreement with that of 1000 genome, GnomAD and ExAC databases. From the reported familial history, the relatedness tests were not felt necessary as the pedigree confirmed correct parenthood for all affected/unaffected samples (Appendix A). From the final list of segregated variants (Table 1), we identified three mutations in *MUC5B, FRG1* and *TAF1B* genes, which we deemed extremely rare variants (Table 2). Furthermore, we also found an *AK9* copy number variant (CNV) which was run through SNPsnap [43] for assessing whether the rare allelic variation was seen as enriched for particular biological annotations (Appendix A). Finally, a set of candidate variants inferred from all samples was checked for validation using Sequenom array/plex (Table 3).

We found two independent lines of evidence from various tools, and reached a consensus in detecting variants from each sample with all filtering steps. For example, the pathogenic mutations contributing to each relevant proband were screened first, wherein three variants, viz., *FRG1* (NC_000004.12:g.189957414T > A), *TAF1B* (NC_000002.12:g.9904885C > T) and *MUC5B* (NC_000011.10:g.1238987C > T), were confirmed through ClinVar and GnomAD. On the other hand, we sought to ask whether or not major differences in functional alleles are seen when compared to unaffected controls. From the MAF cutoff of these rare variants, we observed that these variants came up with other alternative allele burdens not seen in our samples. For example, rs79638064 (C/A) was reflected in SIFT, whereas C/T was seen in our probands, indicating that these could be extremely rare pathogenic conditions.

While the FRG1 mutation happens to be a CNV-associated missense seen in Z15 and Z99 samples, it is further augmented by the fact that the low expression of FRG1 is associated with tumor progression in the colon [47]. The TAF1B mutation is highly associated with colorectal tissues, as we found these stop gain/missense mutations to be highly prolific in some of the aggressive CPC probands, viz., Z15, Z19, Z34, Z42, Z46, Z54, Z62, Z66 and Z74. The TAF1B is known to be the second-largest subunit of the TATA box-binding protein (TBP)-containing promoter selectivity factor TIF-1B/SL1, and is connected with ribosomal transcription [48]. We found that a number of detected variants were associated with colon, rectal and genitourinary tissues. The density of these pouch-related tissues with the presence of missense/stop gain mutations could be attributed to the aggressive CPC type IV.

Dissecting the genetic architecture of a rare disease is certainly an arduous task. Our CPC exome analysis was intended to fill this gap by detecting SNVs and CNVs affecting the focal genes/loci. Assessing these variants in CPC has provided ample evidence of strata coherent to CPC traits. In our study, at least three other genes from trio-exome analysis were reported in colon related ailments, viz., *DLC1, HAVCR1* and *GBA3*, but their allele frequencies were not bona fide or comparable. While we have aimed to characterize the variants by employing different approaches, a polygenic model was assumed. This could be compounded with two assumptions, (a) capturing exomes and identifying deleterious mutations from a high depth of coverage exomes, and (b) identifying large cohorts of mutations that fall in a low depth of coverage exomes. Though we observed both classes of genetic variation contributing to the etiology of the disease, inferring proband–parent trios and detecting de novo and transmitted genetic variants is quite a challenge. By considering extremely rare variants and adopting a strategy of identifying them in the high depth exome, we validated all 16 trios. Nevertheless, we could not compare the detection yield inherent to this spectrum of patients owing to a lack of CPC phenotype and trio-exome studies of similar design. Although previous studies have shown relatively similar methods, they detected medically relevant variants in the majority of the diseased phenotypes [11].

### 3.2. Variants of Unknown Significance

Our findings were in agreement with a large number of reports for rare variants, suggesting that the cumulative contribution of variants across different genes is associated with distinct phenotypes. In addition, an important challenge for researchers and clinicians nowadays in investigating rare disorders involves predicting pathogenicity for VUS. With several guidelines mentioned for predicting the pathogenicity of variants [8,47,48], molecular investigators face a daunting task in considering a rare variant as benign or pathogenic and inferring it to be pathogenic. In explaining the germline/heritability of complex variants based on the rare variant hypothesis, we argue that the extreme rare variants are associated with phenotype sampling [49]. Next, we showed how we can influence and prioritize these extreme rare variants and further propose an optimization procedure to check the variants identified between MAF < 0.01 and MAF 0.01%. To address this, we discarded many variants that had MAF < 0.01 and finally expanded the current annotation and prioritization to accommodate the CPC framework.

In rare diseases where only a minority of the population is affected, or where they are prevalent to a specific geographical location, identifying and considering VUS will require thoughtful consideration. Notable among them, the sequestome (*SQSTM1* or *P62*) gene encodes a multifunctional scaffolding protein involved in multiple cellular processes [50,51] besides showing mitochondrial integrity, import and dynamics as a discriminating autophagy receptor [52]. In addition, *P62*/*SQSTM1* is expressed ubiquitously in various cell types such as cytoplasm, nucleus and lysosomes [53] and is known to be overexpressed in various human genitourinary diseases including colon cancer [54], hepatocellular carcinoma [55] and prostate cancer [56] (Appendix A). While *KCNJ12* was also among the genes harboring bona fide variants, our variant classification did not compel us to consider it as extremely rare. It is known to initiate transcription by RNA polymerase I and acts as a channel for regulatory signals, while *KCNJ12* encodes the ATP-sensitive inward rectifier potassium channel [12] and is subtly associated with the repolarization of channels [57,58]. A list of extremely rare variants in the form of SQSTM1 and *KCNJ12* was also considered for validation. The *SQSTM1* mutation is invariably inherited from the father in the case of the Z12 index case, while being inherited from the mother in the case of Z54. As the aforementioned variants were considered to be extremely rare variants, we also found AK9 (6:109528998..109528999 C|-) to be consistently seen across all probands. This deletion is invariably associated with nucleotide metabolism pathways and maintains the homeostasis of cellular nucleotides [59]. Although AK9 (deletion) is seen phenomenally in all probands, we considered it as a rare variant candidate for validation. However, multiple lines of evidence suggest that apart from VUS, mutations yielding somatic copy number alterations (SCNA) could not be ruled as we found some uncommon missense mutations with MAF < 0.05 in *C7orf57*, *C9orf84*, *ORF5AR1*, *FGFR4*, *HLA-DRB5*, *NOTCH2NLA* and *MUC5B* genes, which could be turn out to be non-pathogenic/causal to CPC.

### 3.3. Role of Hypothetical Genes in CPC Pathogenesis

One of the interesting findings that has emerged from our study is the role of the known unknowns or hypothetical genes, which could be predecessors of non-coding, and hence their establishing roles in diseases such as CPC is limited [60]. Notable among them is the C10orf120 gene, which harbors CTCF binding sites, as these mutations remain undefined for most disease types, including cancer [61]. We observed that there was a significant enrichment of CNVs and indels associated with intestinal/colon-related specific genes as they are widespread in tissues showing chromosomal instability, co-occurring with neighboring chromosomal aberrations, and are frequent in colon, rectum and gastrointestinal tumors but rare in other diseases. We argue that this mutational disruption, associated with CTCF binding sites, could be associated with pathogenesis as it appears to be conserved in a majority of CPC probands (Appendix A). Another orphan ORF, viz., C7orf31, also harbors CTCF binding sites and this, in fact, showed significant enrichment for biological processes associated with regulatory, cellular and metabolic pathways (Table 4). Another C10orf120 has somatic variants subtly contributing to CPC pathogenesis and a maximum number of gains and losses observed in the form of CNVs in *CDC27*, *HLA-DRB5* and *MST1L* genes (Table 2b). Although some of these ORFs’ maternal associations and pathogenicity cannot be ruled out, we construe that there are candidate genes that could be promising biomarkers as precursors of CPC, which is beyond the scope of this canonical hypothesis (Appendix A). In addition, we screened our variants from the Indian Genome Variant Database (IGVDB) and found that they are already reported in the Indian subpopulation [62], and this stratification allowed us to review the patterns influencing common and rare variants. In principle, the rare variants were found to have stronger patterns when compared to the common variants. Thus, there is an inherent need to study the mutations in the known unknown regions which would possibly delve into understanding rare diseases.

### 3.4. RNA-Seq Analysis

To gain insights into the role of lncRNAs, we revisited our hypothesis from our previous study [18] and reconfirmed whether NONHSAT002007 was inferred in WES samples with predictions from the NONCODE database. While we did not find mutations in lncRNAs from trio-exome analyses, we argue that the mutations in essential genes tend to be associated or causal for rare diseases, paving way for driver mutations with the mutations in non-coding genes suppressed for selective pressure. On a different note, we aimed to check whether any of the genes harboring mutations were differentially expressed. To check this, we employed RNA-Seq from the transcriptome pair of CPC type-4 (proband and its unaffected parent), and we observed several transcripts, alternative splice variants and fusion genes. However, none of them could be associated with the causal genes inferred from the exome study (Appendix A). Although we found *RGPD2* and *RGPD4* genes known to be significantly associated with bowel/colon as among the top enriched, nevertheless it is hypothetical to infer global gene expression from just a pair of datasets. This approach, if studied on all samples, we believe, could identify transcripts present at low levels, which in fact could be associated with the pathogenesis of CPC. As the CPC cases emerge, it is difficult to classify the clinical significance of pathogenic variants without the trio analysis, especially the interpretation of de novo variants. The trio exome data could provide insights into whether or not the variant could be inherited, and the carrier mutation could be transcended to the progeny. In conclusion, we argue that the genetic variability in CPC has remarkable significance not only with the parents, but also within each proband. With genetic diseases being leading causes of death in infants, rapid clinical/trio exome sequencing could provide early diagnoses which could make an impact on decision making in critically ill pediatric patients. Although more efficient early diagnosis could be made with interventions using chromosomal microarray (CMA), it was shown that the clinical utility of trio/exome/whole genome sequencing is more than the CMA [63]. The discovery of causal mutations could provide insights into developmental disorders/anorectal malformation such as CPC and its etiology, which closes the gaps of surgery, moving precision therapy forward.

## 4. Conclusions

Our findings confirm de novo heterozygous missense mutations in 16 proband-parent trios, which could provide insights into CPC manifestation and its etiology. This study sheds light on how trio-based WES technologies can play a significant role in the identification of associated/causal mutations for rare diseases. In this work, we identify de novo heterozygous missense mutations in 16 proband-parent trios, and further discover VUS which could provide insights into CPC manifestation and its etiology. Our study confirms candidate mutations in genes, viz., *TAF1B*, *MUC5B* and *FRG1*, cause this developmental disorder. In addition, hypothetical genes including C10orf120 harboring CTCF binding sites were predominant in disease causing variants. A significant enrichment of CNVs and indels associated with colon specific genes was found to be predominant. While the RNA-Seq analysis did not delve into characteristic DEGs, we consider that we need a greater sample size to check the gene expression patterns. Variant validation revealed disease-causing mutations associated with CPC and genitourinary diseases, which could close the gaps of surgery by bringing intervention in therapies.

## Figures and Tables

**Figure 1 children-10-00902-f001:**
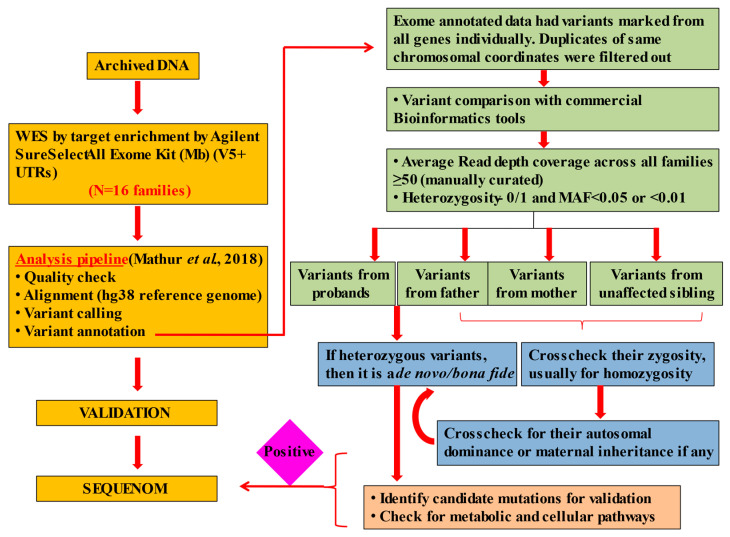
Detailed pipeline for identifying candidate mutations and further screening the data [18].

**Figure 2 children-10-00902-f002:**
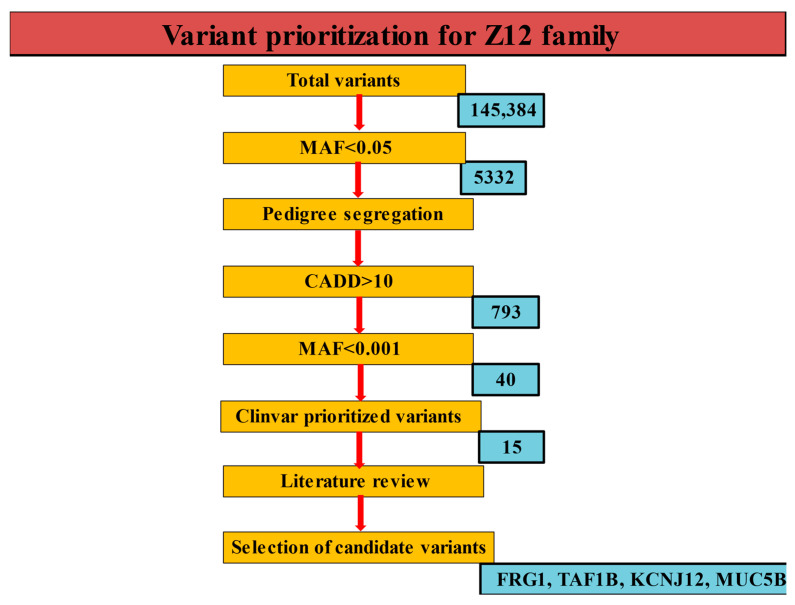
A schematic diagram for the prediction of causal variants in the Z12 family.

**Figure 3 children-10-00902-f003:**
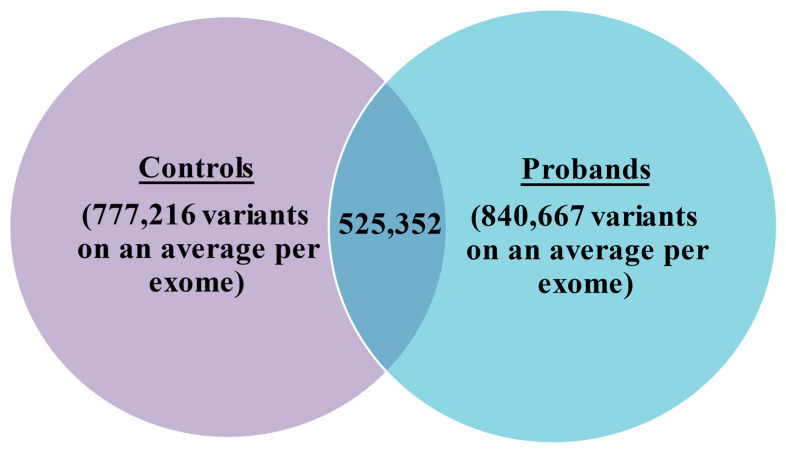
Venn diagram showing variants per exome in controls versus probands. The 525,352 are the common variants between them, which were generated from an average of 840,667 total variants comprising 775,262 SNPs and 65,405 indels per exome in probands as compared to 736,820 SNPs and 63,418 indels in unaffected samples from a total of 777,216 variants.

**Table 1 children-10-00902-t001:** Total variants in cases versus controls.

	Probands (N = 16)	Average Variants for Probands	Controls (N = 46)	Average Variants for Controls	Total Cases	% Probands	% Controls
Total variants	13,450,685	840,667	35,751,971	777,216.76	49,202,656		
SNPs	12,404,198	775,262	33,893,726	736,820.13	46,297,924	25.2104236	68.88596827
Indels	1,046,487	65405.4	2,904,835	63,148.587	3,951,322	2.12689128	5.903817469
Missense	172,232	10764.5	483,790	10,517.174	656,022	0.350046144	0.983259928
Frameshift	10,572	660.75	30,183	656.15217	40,755	0.021486645	0.061344249
Stop gain	2460	153.75	6778	147.34783	9238	0.00499973	0.013775679
Stop lost	469	29.3125	1280	27.826087	1749	0.000953201	0.002601486

**Table 2 children-10-00902-t002:** (a) Extremely rare variants in MUC5B, FRG1 and TAF1B genes, and (b) CNVs associated with variants identified from trio exome.

**(a): Three Extremely Rare Variants in MUC5B, FRG1 and TAF1B Genes**										
**Gene**	**Chromosome Position (hg38) and Human Genome Variation Society (HGVS) Nomenclature**	**Ref/Alt Allele**	**Variant_Annotation**	**MAF/GnomAD**	**Phred_CADD**	**GERP_Score**	**Probands**	**ACMG Criteria**		
FRG1	4-189957414	T/A	Missense	1.9 × 10^−5^	40	4.19	Z19, Z99	PVS1, PM2, PP3		
	NC_000004.12:g.189957414T > A									
TAF1B	2-9904885	C/T	Stop gained	0.0003156	1.31	5.67	Z15, Z19, Z34, Z42, Z46, Z54, Z62, Z66, Z74	PVS1, BS1		
	NC_000002.12:g.9904885C > T,									
MUC5B	11-1238987	C/T	Missense	0.00001972 (rs79638064)	1.07		All other probands except Z12			
	NC_000011.10:g.1238987C > T,									
**(b): CNVs with Gain + Loss Function Associated with Variants Identified from Trio-Exomes**										
**SNP**	**Name**	**Chromosome**	**ChromStart**	**ChromEnd**	**Reference**	**Pubmed**	**Method**	**Sample**	**Gain**	**Loss**
rs1141701	CDC27	chr17	46,006,547	47,199,967	Redon et al., 2006 [44]	17122850	BAC aCGH, SNP array	270	171	43
rs79192142	HLA-DRB5	chr6	32,403,975	32,737,657	Redon et al., 2006 [44]	17122850	BAC aCGH, SNP array	270	165	205
rs79192142	HLA-DRB5	chr6	32,480,351	32,562,509	Coe et al., 2014 [45]	25217958	Oligo aCGH, SNP array	29,084	264	38
rs872964	MST1L	chr1	16,437,837	17,157,486	Redon et al., 2006 [44]	17122850	BAC aCGH, SNP array	270	182	33
rs872964	MST1L	chr1	16,487,425	16,935,752	Vogler et al., 2010 [46]	21179565	Merging, SNP array	1109	115	105
rs872964	MST1L	chr1	16,684,942	16,949,734	Coe et al., 2014 [45]	25217958	Oligo aCGH, SNP array	29,084	163	38

**Table 3 children-10-00902-t003:** Variant validation through sequenom mass array plex.

Rs ID	Chromosomal Position (GRCh38)	Gene	nAllele	Coverage	NoCall	Total	Common	HET	Rage
rs77650227	19:8888842	MUC16	2	100%	0	37	13	4	4
rs10233232	7:48046564	C7orf57	2	100%	0	37	11	9	9
rs358231	4:22818881	GBA3	2	100%	0	37	15	7	7
rs55793208	5:179833099	SQSTM1	2	100%	0	37	9	12	12
rs2947594	10:122697936	C10orf120	2	100%	0	37	5	11	11
rs139094790	8:100709499	PABPC1	2	95%	2	37	12	4	4
rs199887787	5:179837704	SQSTM1	2	100%	0	37	12	4	4
rs2285738	7:25142293	C7orf31	2	100%	0	37	9	12	12
rs1060271	5:179837132	SQSTM1	2	100%	0	37	14	3	3
rs577355457	6:109528998	AK9	2	100%	0	37	25	12	12
rs1612176	17:21416455	KCNJ12	2	100%	0	37	15	5	5
rs6477845	9:111700042	SHOC1	2	100%	0	37	12	12	12

**Table 4 children-10-00902-t004:** *De novo* gene set enrichment analyses.

Name	*p*-Value	Adjusted *p*-Value	Z-Score	Combined Score
Pyrimidine-containing compound transmembrane transport (GO:0072531)	0.003495	0.05015	−3.67	20.77
Regulation of nuclease activity (GO:0032069)	0.003495	0.05015	−2.86	16.16
Adenine nucleotide transport (GO:0051503)	0.003994	0.05015	−2.77	15.32
Purine ribonucleotide transport (GO:0015868)	0.003994	0.05015	−3.08	17.01
Vitamin transmembrane transport (GO:0035461)	0.00499	0.05015	−2.7	14.32
Positive regulation of execution phase of apoptosis (GO:1900119)	0.005488	0.05015	−3.3	17.19
Negative regulation of metabolic process (GO:0009892)	0.006482	0.05015	−1.67	8.4
Negative regulation of oxidoreductase activity (GO:0051354)	0.006482	0.05015	−2.52	12.7
DNA-templated transcriptional preinitiation complex assembly (GO:0070897)	0.00698	0.05015	−1.87	9.28
Regulation of execution phase of apoptosis (GO:1900117)	0.00698	0.05015	−2.52	12.52
Regulation of oxidoreductase activity (GO:0051341)	0.00698	0.05015	−2.24	11.14
Sensory perception of taste (GO:0050909)	0.00698	0.05015	−2.12	10.5
Negative regulation of focal adhesion assembly (GO:0051895)	0.007476	0.05015	−2.66	13.04
Negative regulation of Rho protein signal transduction (GO:0035024)	0.007476	0.05015	−2.58	12.63
Regulation of secretion (GO:0051046)	0.007476	0.05015	−2.06	10.08
Negative regulation of adherens junction organization (GO:1903392)	0.007973	0.05015	−2.57	12.43
rRNA transcription (GO:0009303)	0.008469	0.05015	−1.91	9.09
Negative regulation of stress fiber assembly (GO:0051497)	0.008966	0.05015	−2.22	10.44
Cell-substrate adherens junction assembly (GO:0007045)	0.009462	0.05015	−2.18	10.15
Focal adhesion assembly (GO:0048041)	0.009462	0.05015	−2.19	10.23
Negative regulation of cell junction assembly (GO:1901889)	0.009957	0.05026	−2.36	10.87
Negative regulation of actin filament bundle assembly (GO:0032232)	0.01045	0.05036	−1.99	9.07
Positive regulation of dephosphorylation (GO:0035306)	0.01243	0.05111	−1.53	6.72
Regulation of transmembrane transport (GO:0034762)	0.01243	0.05111	−1.95	8.57
Negative regulation of catalytic activity (GO:0043086)	0.01342	0.05111	−1.52	6.54
Negative regulation of cell-matrix adhesion (GO:0001953)	0.01392	0.05111	−1.73	7.39
Acute inflammatory response (GO:0002526)	0.01441	0.05111	−1.63	6.92
Primary neural tube formation (GO:0014020)	0.01441	0.05111	−2.03	8.59
Transcription elongation from RNA polymerase I promoter (GO:0006362)	0.01441	0.05111	−2.13	9.05
Tube closure (GO:0060606)	0.0149	0.05111	−2.29	9.62
Positive regulation of protein dephosphorylation (GO:0035307)	0.01589	0.05111	−1.83	7.59
Termination of RNA polymerase I transcription (GO:0006363)	0.01589	0.05111	−1.43	5.9
Neural tube closure (GO:0001843)	0.01687	0.05111	−1.81	7.4
Transcription from RNA polymerase I promoter (GO:0006360)	0.01687	0.05111	−1.65	6.73
Transcription initiation from RNA polymerase I promoter (GO:0006361)	0.01687	0.05111	−1.7	6.93
Regulation of ion transmembrane transport (GO:0034765)	0.01786	0.05258	−1.59	6.39
Negative regulation of Ras protein signal transduction (GO:0046580)	0.01884	0.05398	−1.39	5.5
Regulation of RNA metabolic process (GO:0051252)	0.01983	0.05492	−1.37	5.36
Regulation of protein dephosphorylation (GO:0035304)	0.02032	0.05492	−1.94	7.55
Sensory perception of bitter taste (GO:0050913)	0.02081	0.05492	−1.38	5.35
Regulation of multicellular organismal development (GO:2000026)	0.02179	0.05492	−1.95	7.47
Response to hydrogen peroxide (GO:0042542)	0.02179	0.05492	−1.59	6.08
Regulation of focal adhesion assembly (GO:0051893)	0.02228	0.05492	−1.9	7.23
Positive regulation of gene expression, epigenetic (GO:0045815)	0.02375	0.05687	−1.58	5.9
Heart morphogenesis (GO:0003007)	0.02521	0.05687	−1.48	5.45
Negative regulation of viral genome replication (GO:0045071)	0.02521	0.05687	−1.45	5.33
Regulation of ion transport (GO:0043269)	0.02521	0.05687	−1.88	6.92
Negative regulation of cellular catabolic process (GO:0031330)	0.02814	0.05944	−1.2	4.28
Positive regulation of cell death (GO:0010942)	0.02863	0.05944	−1.98	7.02
Regulation of Rho protein signal transduction (GO:0035023)	0.02863	0.05944	−2	7.09
Regulation of stress fiber assembly (GO:0051492)	0.03009	0.05944	−1.28	4.47
Response to reactive oxygen species (GO:0000302)	0.03009	0.05944	−1.26	4.4
Negative regulation of viral life cycle (GO:1903901)	0.03058	0.05944	−1.39	4.86
Regulation of actin filament-based process (GO:0032970)	0.03106	0.05944	−1.25	4.34
Regulation of viral genome replication (GO:0045069)	0.03155	0.05944	−1.32	4.56
Cellular response to type I interferon (GO:0071357)	0.03252	0.05944	−1.35	4.63
O-glycan processing (GO:0016266)	0.03252	0.05944	−1.67	5.71
Type I interferon signaling pathway (GO:0060337)	0.03252	0.05944	−2.23	7.62
Interferon-gamma-mediated signaling pathway (GO:0060333)	0.03495	0.06278	−1.28	4.28
Regulation of gene expression, epigenetic (GO:0040029)	0.03688	0.06471	−1.34	4.41
Regulation of actin cytoskeleton organization (GO:0032956)	0.03737	0.06471	−1.38	4.53
Nucleic acid-templated transcription (GO:0097659)	0.03785	0.06471	−1.75	5.74
Activation of cysteine-type endopeptidase activity involved in apoptotic process (GO:0006919)	0.0393	0.06508	−1.33	4.3
Cation transport (GO:0006812)	0.0393	0.06508	−1.25	4.06
Cell-matrix adhesion (GO:0007160)	0.04459	0.07272	−1.44	4.47
Regulation of cytoskeleton organization (GO:0051493)	0.04699	0.07474	−1.6	4.91
Regulation of cell death (GO:0010941)	0.04747	0.07474	−1.31	4.01
Negative regulation of cell motility (GO:2000146)	0.04794	0.07474	−1.2	3.64
DNA-templated transcription, termination (GO:0006353)	0.05224	0.08013	−1.44	4.26
DNA-templated transcription, elongation (GO:0006354)	0.05319	0.08013	−1.68	4.92
Positive regulation of cysteine-type endopeptidase activity involved in apoptotic process (GO:0043280)	0.05367	0.08013	−1.34	3.93
Protein O-linked glycosylation (GO:0006493)	0.05652	0.08276	−1.24	3.57
Cellular response to interferon-gamma (GO:0071346)	0.057	0.08276	−1.88	5.39
Negative regulation of cell migration (GO:0030336)	0.05937	0.08368	−1.22	3.45
Stimulatory C-type lectin receptor signaling pathway (GO:0002223)	0.05937	0.08368	−1.29	3.64
Innate immune response activating cell surface receptor signaling pathway (GO:0002220)	0.06078	0.08368	−1.4	3.92
Organic anion transport (GO:0015711)	0.06078	0.08368	−1.12	3.13
Regulation of small GTPase mediated signal transduction (GO:0051056)	0.06832	0.09284	−1.25	3.37
Heart development (GO:0007507)	0.07253	0.09732	−1.73	4.54
Positive regulation of protein modification process (GO:0031401)	0.07906	0.1047	−1.02	2.59
DNA-templated transcription, initiation (GO:0006352)	0.08968	0.1169	−1.28	3.08
Receptor-mediated endocytosis (GO:0006898)	0.0906	0.1169	−1.32	3.18
Protein homo-oligomerization (GO:0051260)	0.09152	0.1169	−1.41	3.38
Protein oligomerization (GO:0051259)	0.1038	0.131	−1.63	3.68
Apoptotic process (GO:0006915)	0.1102	0.1374	−1.9	4.2
Sensory perception of smell (GO:0007608)	0.1146	0.1413	−1.57	3.4
Inflammatory response (GO:0006954)	0.1196	0.1457	−1.67	3.54
Endocytosis (GO:0006897)	0.1245	0.1499	−1.59	3.32
Ion transport (GO:0006811)	0.1359	0.1619	−1.09	2.18
Positive regulation of apoptotic process (GO:0043065)	0.1438	0.1693	−1.58	3.06
Regulation of cell migration (GO:0030334)	0.1477	0.172	−1.19	2.27
Sensory perception of chemical stimulus (GO:0007606)	0.1524	0.1756	−1.51	2.83
Transcription, DNA-templated (GO:0006351)	0.1649	0.1879	−1.73	3.12
Negative regulation of cell proliferation (GO:0008285)	0.1678	0.1892	−1.72	3.07
Cellular macromolecule biosynthetic process (GO:0034645)	0.1695	0.1892	−1.63	2.9
Transmembrane transport (GO:0055085)	0.1742	0.1923	−1.28	2.24
Gene expression (GO:0010467)	0.188	0.2054	−1.17	1.96
Regulation of intracellular signal transduction (GO:1902531)	0.1925	0.2082	−1.82	2.99
Neutrophil degranulation (GO:0043312)	0.2157	0.2298	−1.85	2.85
Neutrophil activation involved in immune response (GO:0002283)	0.2173	0.2298	−1.17	1.78
Neutrophil mediated immunity (GO:0002446)	0.2189	0.2298	−1.79	2.72
Positive regulation of cellular process (GO:0048522)	0.2316	0.2407	−1.26	1.84
Negative regulation of cellular process (GO:0048523)	0.2375	0.2445	−1.29	1.85
Cytokine-mediated signaling pathway (GO:0019221)	0.2755	0.2807	−1.25	1.61
Regulation of cell proliferation (GO:0042127)	0.3145	0.3175	−1.05	1.21
Positive regulation of gene expression (GO:0010628)	0.3255	0.3255	−1.52	1.71

## Data Availability

All files are already available with Bioproject id: PRJNA793375 and Biosample id: SUB11892306: https://www.ncbi.nlm.nih.gov/search/all/?term=PRJNA793375 (Accessed on 22 September 2021).

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
