# Peer review of "Whole Exome-Trio Analysis Reveals Rare Variants Associated with Congenital Pouch Colon"

_children, 2023, doi:10.3390/children10050902_

Round 1

Reviewer 1 Report

Dear Authors,

No further comments on your manuscript.

Regards,

Reviewer

Author Response

Thank you very much 

Reviewer 2 Report

Gupta et al., has performed whole exome-trio analysis to identify rare variants associated with congenital pouch colon. The main concern is that the role of the hypothetical genes in CPC pathogenesis is not clear. 

Author Response

Gupta et al., has performed whole exome-trio analysis to identify rare variants associated with congenital pouch colon. The main concern is that the role of the hypothetical genes in CPC pathogenesis is not clear. 

Thank you for your point.  The known unknown regions in the form of hypothetical protein-coding genes are not annotated and we have shown that the mutations in these genes could be detrimental to the phenotype.  We have further elaborated this now with references. Hope this is up to your expectations

Reviewer 3 Report

This article described the genetic analysis of congenital pouch colon(CPC) with WES in an Indian cohort. The authors newly found rare variants, viz. TAF1B, MUC5B, and FRG1. This is a crucial study for solving the mechanism in rare diseases such as CPC, which affects urological matter. However, there are some suggestions about this article. 1. Unfortunately, the sample size was so small to generalize the evidence. 2.   The authors could present the perspective on how to close gaps of surgery, taking forward precision therapy in discussion with this evidence, as the authors stated in the abstract. 

Author Response

This article described the genetic analysis of congenital pouch colon(CPC) with WES in an Indian cohort. The authors newly found rare variants, viz. TAF1B, MUC5B, and FRG1. This is a crucial study for solving the mechanism in rare diseases such as CPC, which affects urological matter. However, there are some suggestions about this article. 

  1. Unfortunately, the sample size was so small to generalize the evidence. 

Thankyou for the comment. Since CPC is a rare disease that occurs congenitally, collecting samples and performing exome trios for a rare disease is very tough. We were also incapacitated by the funding  

  1.   The authors could present the perspective on how to close gaps of surgery, taking forward precision therapy in discussion with this evidence, as the authors stated in the abstract. 

Thank you. We have added the following sentence before conclusions

The discovery of causal mutations could provide insights into the developmental disorders/anorectal malformation such as CPC and its etiology, which closes the gaps of surgery, taking forward to precision therapy.

Reviewer 4 Report

The manuscript reports a WES study performed on 64 samples that included 16 patients affected by Congenital Pouch Colon (CPC).

Although the title reports only the WES analysis, the authors performed also RNA-seq, SNP genotypying and some statistical analysis.

I think that, although the rationale of the research is clear and the amount of patients and data are consistent, the manuscript would need a consistent formatting review in order to better explain the results obtained and the techniques applied.

I will argue by points:

  • Introduction → I will suggest to better balance the part dedicated to the disease and the part dedicated to the applied methods
  • Materials and Methods

- paragraph 2.2 → it is not clear the coverage of the sequencing. In line 100 the author specify a coverage 110X but in supplementary table 1, the average sequencing depth is around 10. Is this the same metric? Please clarify

Moreover I think that all this paragraph needs to be rephrase: it is not clear which is the followed process and what “manually” means. They also used “awk” and “grep” that are two very common commands which I don't think should be written in a manuscript. Thus, it is notclear the MAF adopted for filtering variants: why the authors say MAF<0.001 and MAF<=0.01% are they both percentage or not? (the same in line 240)

- paragraph 2.3 → the author spent many lines to explain what is commonly defined with the term "segregation". I don't think it's necessary. Otherwise, specify why

In general, I had a great difficulty to follow the pipeline: only denovo and recessive variants were considered? What does the authors mean with “enrichment analysis”?please better explain all the passagges.

- paragraph 2.4 → The RNA sequencing it is cited for the first time. The authors do not explain why the RNA-seq was done nor on which samples it was done (cells? samples? Why?)

- paragraph 2.5 → the authors adopted the Sequenom genotypying but it is not clear if it is to validate variants or for another aim. Moreover, the authors must specify why they included only 37 DNA samples

  • Results and Discussion: this paragraph is very confused and difficult to read. There are some superfluous information such as, for example, the total number of variants and figure1 (what is it for?) while many information are missing (eg. where the CNV come from? Which bioinformatic tool has been used? On wes data?). I cannot see table 1 and table 2 with the selected mutations.

-lines 198-207: it is not clear where the data come from. How can the association be evaluated? I am confused

- from lines 210: this paragraph needs to be rephrase and move to the discussion session

It is very difficult to follow the reasoning and I would like to suggest to modify the results session in order to be more orderly, including the use of RNA-seq and the validation of variants

I would like also to add some minor points which:

- I cannot see the tables of the manuscript. I only see supplementary tables. Please check and verify

- ref. 1: it is the reference of the tool SIFT for the pathogenicity prediction. Are the authors sure that this is the right reference for the WES application?

- Line 46-48, ref 5-7: the authors cite some papers published 10 years ago. Can they be replaced by more recent works?

- Line 54-69: the authors spent 15 lines in order to do examples regarding the application of trio WES in resolving mendelian cases. Are these citations necessary? Are they exhaustive?

- Line 70: please add the incidence of the CPC disease

- Line 91: why the authors use the word “restricted”? Please clarify or rephrase

- Line 115: please add the bioinformatic tools used

- paragraph 3.1: please write the genomic mutation in a more appropriate way (eh. ChrN:position)

- conclusion: please reconsider some paragraphs of the findings to be moved to this session

Author Response

The manuscript reports a WES study performed on 64 samples that included 16 patients affected by Congenital Pouch Colon (CPC).

Although the title reports only the WES analysis, the authors also performed RNA-seq, SNP genotypying and some statistical analysis.

 I think that, although the rationale of the research is clear and the amount of patients and data are consistent, the manuscript would need a consistent formatting review in order to better explain the results obtained and the techniques applied.

Thank you for your comments and suggestions, constructive criticism 

I will argue by points:

    • Introduction → I will suggest to better balance the part dedicated to the disease and the part dedicated to the applied methods
  • Thankyou for the comment. We edited the Introduction and balanced both the disease section and applied methods in it.
  •  
  • Materials and Methods

- paragraph 2.2 → it is not clear the coverage of the sequencing. In line 100 the author specify a coverage 110X but in supplementary table 1, the average sequencing depth is around 10. Is this the same metric? Please clarify

 Thank you. Although the depth of coverage is aimed at 110x whence sequencing , we have screened those variants that are minimum depth of 10 whilst filtering. 

Moreover I think that all this paragraph needs to be rephrase: it is not clear which is the followed process and what “manually” means. They also used “awk” and “grep” that are two very common commands which I don't think should be written in a manuscript. Thus, it is not clear the MAF adopted for filtering variants: why the authors say MAF<0.001 and MAF<=0.01% are they both percentage or not? (the same in line 240)

Thankyou for the comment. We removed the word manually from the methods section.

The One with % is 0.01 % which is 0.001 which is extremely rare variant as indicated. 

- paragraph 2.3 → the author spent many lines to explain what is commonly defined with the term "segregation". I don't think it's necessary. Otherwise, specify why

Thankyou for the comment. “Segregation” is used to report the variants extracted from whole exome trio analysis. We have changed the term in the article.

In general, I had great difficulty following the pipeline: only denovo and recessive variants were considered? What does the author mean with “enrichment analysis”?please better explain all the passages.

Thankyou for the comment. By enrichment analysis we observed the role of variants in CPC pathogenesis. We observed an enrichment of CNVs and indels associated with intestinal/colon related-specific genes.

- paragraph 2.4 → The RNA sequencing it is cited for the first time. The authors do not explain why the RNA-seq was done nor on which samples it was done (cells? samples? Why?)

Thankyou for the comment. RNA sequencing was done on CPC-type 4 samples. To gain insights into the differentially expressed genes, RNA sequencing was done. We have added a section of differential expressed genes as 3.4 in results and discussion.     

- paragraph 2.5 → the authors adopted the Sequenom genotypying but it is not clear if it is to validate variants or for another aim. Moreover, the authors must specify why they included only 37 DNA samples

Thankyou for the comment. Sequenom genotyping was done to validate the variants on a multiplexed iPLEX assay. Only 37 DNA samples passed the quality check thus were included in Sequenom analysis.

  • Results and Discussion: this paragraph is very confused and difficult to read. There are some superfluous information such as, for example, the total number of variants and figure1 (what is it for?) while many information are missing (eg. where the CNV come from? Which bioinformatic tool has been used? On wes data?). I cannot see table 1 and table 2 with the selected mutations.

Thankyou for the comment. Figure 1 represents a venn diagram depicting the total number of variants with total SNPs and indels.

CNVs were determined using SNPsnap bioinformatic tool. This information is included in section 3.1 of results and discussion.

Table 1 and 2 are included in the manuscript.

-lines 198-207: it is not clear where the data come from. How can the association be evaluated? I am confused

Sorry for the confusion, what we aimed for was to check whether or not the genes harboring mutations are differentially expressed as well from our erstwhile RNA-Seq data.  We have paraphrased the sentence now

On a different note, we aimed to check if any of the genes harboring mutations were differentially expressed.   To check this, we employed RNA-Seq from the transcriptome pair of CPC type-4 (proband and its unaffected parent), and we observed several transcripts, alternative splice variants and fusion genes but none of them could be associated with the causal genes inferred from exome study (Supplementary Table 5). 

- from lines 210: this paragraph needs to be rephrase and move to the discussion session

It is very difficult to follow the reasoning and I would like to suggest to modify the results session in order to be more orderly, including the use of RNA-seq and the validation of variants

 Thank you. We have paraphrased sentences and moved and cleared sections. Please check 

  I would like also to add some minor points which:

- I cannot see the tables of the manuscript. I only see supplementary tables. Please check and verify

Thank you for the comment. Tables are added in the manuscript.

- ref. 1: it is the reference of the tool SIFT for the pathogenicity prediction. Are the authors sure that this is the right reference for the WES application?

Thank you for the comment. Reference 1 has been changed for WES application.

- Line 46-48, ref 5-7: the authors cite some papers published 10 years ago. Can they be replaced by more recent works?

Thank you. Recent references have been added.

- Line 54-69: the authors spent 15 lines in order to do examples regarding the application of trio WES in resolving mendelian cases. Are these citations necessary? Are they exhaustive?

Thank you. To get more insights into the trio-based whole exome sequencing for rare diseases, a few more examples are now cited in the introduction section.  

- Line 70: please add the incidence of the CPC disease

Thank you. Added incidence of CPC in the introduction section.

- Line 91: why the authors use the word “restricted”? Please clarify or rephrase

Thank you. We rephrased the sentence.

- Line 115: please add the bioinformatic tools used

Thank you.  Bioinformatic tools are now added. 

- paragraph 3.1: please write the genomic mutation in a more appropriate way (eh. ChrN:position)

Thank you. The genomic mutations are written in appropriate manner.

- conclusion: please reconsider some paragraphs of the findings to be moved to this session

We tried to paraphrase conclusions now. Kindly check 

Round 2

Reviewer 2 Report

Thank you for addressing my comments.

I have no additional comments for authors.

Author Response

Thank you very much for all the suggestions rendered. 

Reviewer 4 Report

Thank you to the authors for their reply to my comments. I have some other suggestions.

1. I have a pdf file with no line numbers and I would suggest always including numbers in order to help reviewers to better identify changes. 

2. I asked many questions to the authors but not all the comments seem to be added to the manuscript. I thanks the authors for replyning to me, but I think that they need to improve also the manuscript in order to facilitated the reading.

3. I suggest again removing the words "grep" and "awk" from the manuscript. It sounds like a "find" in a doc file and I think it is not appropriate (please clarify if I don't well understand)

4. Please, better write the MAF you used for filtering. once it is written as an absolute number (0.001), and once as a percentage. It is not clear. Moreover please specify why the first filter was set to 0.05 and the second one to 0.001. Are there any reasons? otherwise, please fix what is written in the workflow.

5. can you check if the word "although" in "material and method" is the write one? it seems incorrect

6. please write in the introduction/abstract the use of RNA-seq

7. I think that by studying a WES it could be possible to wont have lncRNA in the design. Can you please specify if the library also included those?

8. Please write all the variants according to HGVS guide lines

Author Response

Responses

We thank the reviewers for kind inputs and suggestions. We have further incorporated your suggestions and line by line response is accorded here with.

Thank you to the authors for their reply to my comments. I have some other suggestions.

  1. I have a pdf file with no line numbers and I would suggest always including numbers in order to help reviewers to better identify changes. 

Thank you. We have now inserted line numbers.  Apologies, the first version has had line numbers

  1. I asked many questions to the authors but not all the comments seem to be added to the manuscript. I thanks the authors for replyning to me, but I think that they need to improve also the manuscript in order to facilitated the reading.

 Thank you. All the changes were highlighted in colors. We request you to check again please.  We have further made some more changes. 

  1. I suggest again removing the words "grep" and "awk" from the manuscript. It sounds like a "find" in a doc file and I think it is not appropriate (please clarify if I don't well understand)

Thank you. We have removed it. The other reviewer in round 1 asked us how we parsed.  Now we revert back to that  

  1. Please, better write the MAF you used for filtering. once it is written as an absolute number (0.001), and once as a percentage. It is not clear. Moreover please specify why the first filter was set to 0.05 and the second one to 0.001. Are there any reasons? otherwise, please fix what is written in the workflow.

 Thank you for these excellent comments. From our pipeline, we sought to take MAF <0.05 so as not to miss any variants as per standard protocol. However, as it is a rare disease, we have screened mutations with MAF <=0.01 and also we found extremely rare variants with MAF <0.01% which is 0.001 and hence we have mentioned that in the variant type next to the gene

  1. can you check if the word "although" in "material and method" is the write one? it seems incorrect

Thank you. We have removed the preceding sentence. It reads as follows

We confined our pool of analyses to all probands (11 male and 5 female) and unaffected parents/siblings.

  1. please write in the introduction/abstract the use of RNA-seq

 Thank you. We have added the following in abstract

The  proband exomes were compared with the unaffected sibling/family members and we sought to ask whether any variants of significant interest are associated with the CPC manifestation.  The WES data from a total of 64 samples including 16 affected neonates (11 male and 5 female) with their parents and unaffected siblings were used for the study.  We examined the role of rare allelic variation associated with CPC in a 16 proband/parent trio family comparing the mutations to that of their unaffected parents/siblings. We also have performed RNA-Seq as a pilot to find whether or not the genes harboring these mutations are differentially expressed.  Our study revealed extremely rare variants, viz. TAF1B, MUC5B  and FRG1 which were further validated for disease-causing mutations associated with CPC further closing the gaps of surgery by bringing intervention in therapies.

And conclusions as follows

Our findings confirm de novo heterozygous missense mutations in 16 proband-parent trios which could provide insights into CPC manifestation and its etiology. This study sheds light on how trio‐based WES technologies can play a significant role in identification of associated/causal mutations for rare diseases.  In this work, we identify de novo heterozygous missense mutations in 16 proband-parent trios and further discover VUS which could provide insights into CPC manifestation and its etiology. Our study confirms candidate mutations in genes, viz. TAF1B, MUC5B and FRG1 cause this developmental disorder. In addition, hypothetical gene including C10orf120 harboring CTCF binding sites was predominant as disease causing variant. A significant enrichment of CNVs and indels associated with colon specific genes were found predominant. While the RNA-Seq analysis didn’t delve upon characteristic DEGs, we contemplate that we need more sample size to check the gene expression patterns. Variant validation revealed disease causing mutations associated with CPC and genitourinary diseases which could close the gaps of surgery by bringing intervention in therapies.

  1. I think that by studying a WES it could be possible to wont have lncRNA in the design. Can you please specify if the library also included those?

Thank you. The library prep was Agilent SureSelect V4+5’ UTR and this is where we have earlier found a lncRNA: https://www.mdpi.com/2218-273X/8/3/95   The untranslated regions harbor variants sitting in non-coding regions ( intergenic regions) and many other groups have also found such candidates. 

  1. Please write all the variants according to HGVS guide lines

Thank you, all the protein-coding genes and variants enlisted were put up in HGVS and HUGO nomenclature:  For example, the pathogenic mutations contributing to each relevant proband were screened first wherein three variants, viz. FRG1 (NC_000004.12:g.189957414T>A), TAF1B (NC_000002.12:g.9904885C>T) and MUC5B (NC_000011.10:g.1238987C>T) confirmed through ClinVar and GnomAD. 

Also seen in Table 1